# Investigation and Mitigation of Noise Contributions in a Compact Heterodyne Interferometer

**DOI:** 10.3390/s21175788

**Published:** 2021-08-28

**Authors:** Yanqi Zhang, Adam S. Hines, Guillermo Valdes, Felipe Guzman

**Affiliations:** 1Department of Aerospace Engineering, Texas A&M University, 701 H.R. Bright Bldg., College Station, TX 77843, USA; yqzhang@tamu.edu (Y.Z.); adamhines@tamu.edu (A.S.H.); gvaldes@tamu.edu (G.V.); 2Wyant College of Optical Sciences, The University of Arizona, 1630 E. University Blvd., Tucson, AZ 85721, USA

**Keywords:** heterodyne laser interferometer, displacement measuring interferometry (DMI), inertial sensing, noise subtraction

## Abstract

We present a noise estimation and subtraction algorithm capable of increasing the sensitivity of heterodyne laser interferometers by one order of magnitude. The heterodyne interferometer is specially designed for dynamic measurements of a test mass in the application of sub-Hz inertial sensing. A noise floor of $3.31\times 10^{-11}$
m/$\sqrt{\mathrm{Hz}}$ at 100 mHz is achieved after applying our noise subtraction algorithm to a benchtop prototype interferometer that showed a noise level of $2.76\times 10^{-10}$
m/$\sqrt{\mathrm{Hz}}$ at 100 mHz when tested in vacuum at levels of $3\times 10^{-5}$ Torr. Based on the previous results, we investigated noise estimation and subtraction techniques of non-linear optical pathlength noise, laser frequency noise, and temperature fluctuations in heterodyne laser interferometers. For each noise source, we identified its contribution and removed it from the measurement by linear fitting or a spectral analysis algorithm. The noise correction algorithm we present in this article can be generally applied to heterodyne laser interferometers.

## 1. Introduction

In the past decades, displacement measuring interferometry (DMI) has extended its application to gravitational wave (GW) detection, including free-falling test mass measurements in space-based gravitational wave detection as the Laser Interferometer Space Antenna (LISA) and its technology demonstrator, LISA Pathfinder [1,2,3,4,5], intersatellite displacement measurements as in the mission GRACE Follow-On that utilizes a Laser Ranging Interferometer [6], and inertial sensor development for seismic activity monitoring in ground-based Laser Interferometer Gravitational-Wave Observatory (LIGO) [7,8]. The above applications require high sensitivity or, in other words, a low noise floor in the frequency regime between tens of micro-Hz to a few Hz. The test mass motion is usually expected to range from a few microns to several millimeters, where a DMI with a large dynamic range is crucial to avoid phase readout ambiguities. Among various DMI techniques, heterodyne laser interferometry has the advantages of multi-fringe measurement range, inherent directional sensitivity, and fast detection speed. The development of a common-mode rejection scheme in heterodyne interferometer [1,9] provides a promising solution to displacement metrology applications due to the high rejection ratio to common-mode noise sources. Further improvements on the interferometer performance towards picometer level sensitivity require the knowledge of the residual noise sources and the contribution of each noise source.

Efforts have been made to identify noise sources in heterodyne interferometry [10,11,12,13,14,15,16,17,18,19,20,21,22,23,24]. The inherent periodic error in heterodyne interferometry is mitigated by spatially separating the beams to reduce frequency and polarization mixing [10,11]. The effect of non-linear optical path difference (OPD) noise caused by electric sideband interference is investigated in [12,13], where its origin and mitigation method by active feedback control are introduced. The laser frequency, as the “ruler” of interferometric measurements, has been studied in terms of noise behaviors [14] and various stabilization methods [15,16,17,18]. Photoreceivers are characterized in [19], where the predominant noise sources are identified, and a photodiode with low current noise is designed. The effects of temperature fluctuations in a low frequency regime are analyzed in [20,21]. Other common noise sources in DMI, such as environmental noises and phasemeter noises, are also identified in previous work [22,23,24]. However, an inclusive post-processing algorithm to characterize and correct multiple noise sources in the system has not yet been published.

In this article, we present a noise identification and mitigation algorithm that can be applied to general heterodyne interferometers. We have tested this in a compact laboratory benchtop interferometer. In Section 2, we introduce the interferometer design, which is built to have a high common-mode rejection ratio and be periodic-error free. In Section 3, we analyze residual noise sources based on the differential measurement results. The contributions from distinct noise sources including the non-linear OPD noise, laser frequency noise, and temperature fluctuations, are identified and subtracted from the differential measurement. The noise correction results show a noise floor of $3.31\times 10^{-11}$ m/$\sqrt{\mathrm{Hz}}$ at 100 mHz, which is enhanced by an order of magnitude from the original differential measurement, reaching the detection system noise limit above 1 Hz.

## 2. Compact Heterodyne Laser Interferometer

### 2.1. Design and Benchtop Prototype

Figure 1a shows the layout of the compact interferometer design. After passing through two individual acousto-optical frequency shifters (AOFS), two laser beams are shifted by frequencies $\delta f_{1}$ and $\delta f_{2}$, respectively. The heterodyne frequency $f_{\mathrm{het}}$ is the difference between the two laser frequencies, where $f_{\mathrm{het}}=\delta f_{1}-\delta f_{2}$. The two beams enter a 50/50 lateral beam splitter (LBS), generating four parallel beams. All four beams propagate through a polarizing beam splitter (PBS) and a quarter-wave plate (QWP) towards the mirrors on the measurement end. They are then reflected back into the PBS, exiting along an orthogonal direction. In the second LBS, two beam pairs interfere and are detected by photodetectors PD${}_{R}$ and PD${}_{M}$, respectively. The electric fields of the four beams at the interferometer output are given by
(1)$$\begin{array}{cc} E_{1} & =E_{0}\exp\left[i2\pi \left(f+\delta f_{1}\right)t+\phi_{n}\left(0,\delta_{y}\right)+\Delta \phi_{M}\right], \end{array}$$
(2)$$\begin{array}{cc} E_{2} & =E_{0}\exp\left[i2\pi \left(f+\delta f_{1}\right)t+\phi_{n}\left(\delta_{x},\delta_{y}\right)+\Delta \phi_{R}\right], \end{array}$$
(3)$$\begin{array}{cc} E_{3} & =E_{0}\exp\left[i2\pi \left(f+\delta f_{2}\right)t+\phi_{n}\left(0,0\right)\right], \end{array}$$
(4)$$\begin{array}{cc} E_{4} & =E_{0}\exp\left[i2\pi \left(f+\delta f_{2}\right)t+\phi_{n}\left(\delta_{x},0\right)\right], \end{array}$$
where $E_{0}$ is the nominal amplitude of the electric field. The term $\phi_{n}\left(x,y\right)$ denotes the structual noises that imprints on the beam optical pathlengths while propagating through the optical components from the entry plane to the plane defined by the mirror *M*. This noise term $\phi_{n}$ is dependent on the beam path (optical axis) position, which is represented by a coordinate (x,y) in the plane perpendicular to the beam entry directions, as shown in Figure 1a. The phase terms $\Delta \phi_{M}$ and $\Delta \phi_{R}$ represent the additional contributions from the optical pathlength differences between mirrors *M* and $M_{M}$, as well as between *M* and $M_{R}$. As depicted in Figure 1b, two beams with the heterodyne frequency difference ($E_{1}$ and $E_{3}$, $E_{2}$ and $E_{4}$) are combined into one beam pair.

Figure 1b shows a top view of the interferometer. Optical components can be bonded or cemented together to reduce the overall footprint, increase the system stability, and simplify the alignment process. In this design, two interferometers are constructed: (a) the measurement interferometer (MIFO) that measures the OPD between the target mirror M${}_{M}$ and the fixed mirror M, where $\phi_{M}=\phi_{n}\left(0,\delta_{y}\right)-\phi_{n}\left(0,0\right)+\Delta \phi_{M}=\phi_{n}\left(0,\delta_{y}\right)+\Delta \phi_{M}$; (b) the reference interferometer (RIFO) that measures the OPD between the reference mirror M${}_{R}$ and the fixed mirror M, where $\phi_{R}=\phi_{n}\left(\delta_{x},\delta_{y}\right)-\phi_{n}\left(\delta_{x},0\right)+\Delta \phi_{R}=\phi_{n}\left(0,\delta_{y}\right)+\Delta \phi_{R}$. The mirror M is affixed to the same base plate with the interferometer to provide a common reference for MIFO and RIFO. The mirror M${}_{R}$ is mounted as closely as possible to M${}_{M}$ to provide a local reference for target displacement. The irradiance signal detected by the photodetectors PD${}_{R}$ and PD${}_{M}$ are described by
(5)$$\begin{array}{cc} I_{R} & =I_{R0}+A\cos\left(2\pi f_{\mathrm{het}}t+\phi_{n}\left(0,\delta_{y}\right)+\Delta \phi_{R}\right), \end{array}$$
(6)$$\begin{array}{cc} I_{M} & =I_{M0}+B\cos\left(2\pi f_{\mathrm{het}}t+\phi_{n}\left(0,\delta_{y}\right)+\Delta \phi_{M}\right). \end{array}$$

The phases $\phi_{M}$ and $\phi_{R}$ can be extracted by various methods such as phase-locked loops (PLL) [25] or discrete Fourier transforms [1]. By taking the difference between individual phase readouts, the target displacement *d* is calculated by
(7)$$\begin{array}{cc} d & =\frac{\Delta \phi }{2\pi \;\cdot \;2}\lambda =\frac{\phi_{R}-\phi_{M}}{2\pi \;\cdot \;2}\lambda =\frac{\Delta \phi_{R}-\Delta \phi_{M}}{2\pi \;\cdot \;2}\lambda , \end{array}$$
where Δϕ is the differential phase readout, and λ is the source wavelength. This is referred to as the common-mode rejection scheme, where the noises in the common-mode optical paths $\phi_{n}\left(0,\delta_{y}\right)$ are canceled with the presence of a common reference provided by mirror M.

We built a benchtop prototype with commercial optical components and mechanical mounting parts, as shown in Figure 2. Optical components are clamped together mechanically in this assembly to demonstrate the design concept. The laser source is a tunable diode laser (NP Photonics) operating at a nominal wavelength of 1064 nm. Two AOFS (G&H T-M150-0.4C2G-3-F2P) shift the frequencies upwards by 150 MHz and 145 MHz, respectively, in two paths, generating a 5 MHz heterodyne frequency. A commercial phasemeter (Liquid Instruments Moku:Lab) is used to extract the phase from the heterodyne signal using a PLL algorithm with a sampling frequency of 30.5 Hz. All fibers in the system are polarization-maintaining fibers (PMF) to preserve the interference visibility.

### 2.2. Operation Environments

The benchtop system is tested in the vacuum chamber with a pressure of $3\times 10^{-5}$ Torr to reduce the effects from refractive index fluctuation, acoustic noises, and air turbulence. The laser source is coupled into the system by a fiber feedthrough port, while the optical signal is detected after transmitting through an optical window port. Figure 3 shows the operation environment for the preliminary test. A pendulum platform consisting of a breadboard and four stainless steel wires is set inside the chamber for vibration isolation above its resonance. The first resonance mode of the pendulum platform has a frequency of 305 mHz measured by a ring-down test. Viton strips are inserted between the frame of the pendulum platform and the base of the vacuum chamber to mitigate thermal conduction. In addition, a thermal insulation box is built outside the chamber with Styrofoam to reduce thermal coupling from the ambient environment.

### 2.3. Preliminary Test

To evaluate the noise floor of the interferometer, we utilized a single static mirror instead of using individual fixed, reference, and test mirrors. Figure 4 shows the measurement results for MIFO and RIFO, as well as their difference, which represents the sensitivity level of the overall interferometer. An eight-hour measurement is taken in the operation environment described in Section 2.2. Figure 4a, the logarithmic average of the linear spectral density (LSD) plot, shows that the individual interferometers MIFO and RIFO achieve a sensitivity level of $1.64\times 10^{-7}$ m/$\sqrt{\mathrm{Hz}}$ at 100 mHz. The traces of MIFO and RIFO overlap highly due to the common paths shared between the two interferometers. With the common-mode rejection scheme, the sensitivity is enhanced to the level of $2.76\times 10^{-10}$ m/$\sqrt{\mathrm{Hz}}$ at 100 mHz. Figure 4b shows the time series of the measured displacement for the individual interferometers and the differential measurement, which is measured to be $7.56\times 10^{-10}$ m over the duration of 12 min.

## 3. Noise Source Characterization and Suppression

The preliminary test demonstrated that the benchtop compact interferometer reduces the interferometer noise floor by rejecting common-mode pathlength noises, thus enhancing the sensitivity level by three orders of magnitude. The residual noise sources in the differential measurement are investigated in this section to determine their contribution to the overall noise floor. The interferometer performance is further improved by combining a linear fit method [26,27,28] with a spectral analysis method.

### 3.1. Non-Linear OPD Noise

The noise investigation of the LISA Pathfinder interferometer demonstrated that the electromagnetic coupling of the radio-frequency (RF) signals between two AOFS drivers, generates sidebands that imprint on the optical signals [12,13]. Figure 5 shows the measured RF driving signals applied on both AOFS. The sidebands interfere with the RF signals at nominal shifting frequencies, and lead to a heterodyne signal with the frequency equal to the interferometer’s main heterodyne frequency $f_{\mathrm{het}}$. Therefore, this “ghost” signal is detected by the photodetector (PD) and mixed with the actual displacement readout, resulting in a non-linear noise that is dependent on the individual phase readouts $\phi_{M}$ and $\phi_{R}$.

Based on the analysis presented in references [12,13], the non-linear OPD noise caused by the sideband interference can be expressed as
(8)$$\begin{array}{cc} \delta \phi_{\mathrm{OPD}}=\left(C_{1}\cos\left(\frac{\phi_{M}+\phi_{R}}{2}\right)+C_{2}\sin\left(\frac{\phi_{M}+\phi_{R}}{2}\right)\right) & \;\cdot \;\sin\left(\frac{\phi_{M}-\phi_{R}}{2}\right)\\ +\left(C_{3}\cos\left(\phi_{M}+\phi_{R}\right)+C_{4}\sin\left(\phi_{M}+\phi_{R}\right)\right) & \;\cdot \;\sin\left(\phi_{M}-\phi_{R}\right). \end{array}$$

Equation (Equation 8) can be reformatted as
(9)$$\begin{array}{cc} \delta \phi_{\mathrm{OPD}} & =C\;\cdot \;N(\phi_{M},\phi_{R})\\ \; & =\left[C_{1},C_{2},C_{3},C_{4}\right]\;\cdot \;\left[\begin{array}{c} \cos\left(\frac{\phi_{M}+\phi_{R}}{2}\right)\;\cdot \;\sin\left(\frac{\phi_{M}-\phi_{R}}{2}\right)\\ \sin\left(\frac{\phi_{M}+\phi_{R}}{2}\right)\;\cdot \;\sin\left(\frac{\phi_{M}-\phi_{R}}{2}\right)\\ \cos\left(\phi_{M}+\phi_{R}\right)\;\cdot \;\sin\left(\phi_{M}-\phi_{R}\right)\\ \sin\left(\phi_{M}+\phi_{R}\right)\;\cdot \;\sin\left(\phi_{M}-\phi_{R}\right) \end{array}\right], \end{array}$$
where C is the coupling coefficient vector, and N is the non-linear OPD term vector in the phase measurement. A linear fit is then performed between the differential phase Δϕ and the non-linear OPD terms to estimate the coupling factor $\tilde{C}$. The contribution of the non-linear OPD noise is removed, following the equation
(10)$$\Delta \phi_{\mathrm{OPDcorr}}=\Delta \phi -\delta {\tilde{\phi }}_{\mathrm{OPD}}=\Delta \phi -\tilde{C}\;\cdot \;N\left(\phi_{M},\phi_{R}\right).$$

Figure 6 shows the LSD for the original differential measurement results and the results after subtracting the non-linear OPD noise. Applying this noise correction leads to a reduction in the noise floor from $2.76\times 10^{-10}$ m/$\sqrt{\mathrm{Hz}}$ to $3.86\times 10^{-11}$ m/$\sqrt{\mathrm{Hz}}$ at 100 mHz. Table 1 lists the fitted coefficients C and estimated errors δC during the linear fit process.

### 3.2. Laser Frequency Noise

The traceability of interferometric measurements is dependent on the accurate knowledge of the laser source frequency. Therefore, fluctuations of the laser frequency in an interferometer with unequal armlengths directly impact the interferometer phase following the equation
(11)$$\delta \phi_{\mathrm{freq}}=\frac{2\pi \Delta L}{c}\delta \nu ,$$
where ΔL is the pathlength difference between two arms of the interferometer, and δν is the laser frequency fluctuation. For the proposed interferometer, the phase readout is the differential measurement of two individual interferometers MIFO and RIFO. The contribution of laser frequency noise to the measurement can be expressed by
(12)$$\delta \phi_{\mathrm{freq}}=\delta \phi_{\mathrm{fM}}-\delta \phi_{\mathrm{fR}}=\frac{2\pi \delta \nu }{c}\left[\left(L_{M1}-L_{R1}\right)+\left(L_{M2}-L_{R2}\right)\right],$$
where $L_{\mathrm{Mi}}$ is the length of the $i_{\mathrm{th}}$ arm in MIFO, and $L_{\mathrm{Ri}}$ is the length of the $i_{\mathrm{th}}$ arm in RIFO. For the nominal design, $L_{M1}$ is equal to $L_{R1}$ due to the common reference surface provided by the fixed mirror *M*. Therefore, the magnitude of the laser frequency noise is determined by the axial distance between the reference mirror $M_{R}$ and the measurement mirror $M_{M}$ that is mounted on the test mass. In the preliminary test, only one static mirror is used, which means both terms $\left(L_{M1}-L_{R1}\right)$ and $\left(L_{M2}-L_{R2}\right)$ are nominally canceled. However, residual arm length differences remain due to the manufacturing tolerance as well as the imperfect alignment. To evaluate the magnitude of the residual laser frequency noise in our benchtop system, a fiber-based delay-line interferometer (DIFO) is built and integrated with the prototype interferometer as shown in Figure 7.

The DIFO is a heterodyne interferometer with an intentionally designed unequal arm length using a 2-meter fiber in one interferometer arm. The laser frequency fluctuation then gets amplified when coupled into the DIFO phase readout $\phi_{d}$ according to Equation (Equation 12).

Figure 8 shows the linear spectra of the interferometers when injecting a 2 Hz modulation to the laser frequency. The spectrum of DIFO at 2 Hz is $4.82\times 10^{-7}$ m, while the spectrum of RIFO is $5.30\times 10^{-10}$ m. It can also be noted from Figure 8 that, aside from the injected laser frequency modulation peak and its harmonics, the noise floor of DIFO is almost identical to RIFO due to predominantly common environmental noises. This makes it difficult to extract any excess laser frequency noise from the DIFO readout. Therefore, we first subtract the RIFO phase $\phi_{R}$ from DIFO phase $\phi_{D}$, to mitigate the common path noise effects. The residual phase, $\phi_{D}^{\prime }=\phi_{D}-\phi_{M}$, is used to estimate the noise coupling factor. Moreover, the residual phase is band-passed in the frequency regime between 1 mHz and 1 Hz to mitigate temperature fluctuation and mechanical vibration effects.

We perform a linear fit of the band-passed phase, $\phi_{D}^{\prime }$, to the differential phase, to estimate the coupling coefficient $\tilde{K}$. The noise correction procedure, which is applied to the original data, can be described as
(13)$$\Delta \phi_{\mathrm{freqcorr}}=\Delta \phi -\delta {\tilde{\phi }}_{\mathrm{freq}}=\Delta \phi -\tilde{K}\;\cdot \;\phi_{D}^{\prime }.$$

Figure 9 shows the results of identifying the laser frequency noise contribution based on the OPD-noise-corrected differential phase described in Section 3.1. The laser frequency contribution is identified to be $2.28\times 10^{-12}$ m/$\sqrt{\mathrm{Hz}}$ at 100 mHz in this case, and is not the dominant noise source in this measurement. We injected a 2 Hz modulation to the laser frequency in order to independently determine its coupling factor and the effective optical arm length mismatch between the interferometers. The results are shown in Figure 8, and we obtain an effective arm length mismatch, ΔL, of $2.31\times 10^{-5}$ m. From Equation (Equation 11), it is inferred that the effects of laser frequency fluctuations are suppressed by a small ΔL, and therefore is not a limiting factor for the current experimental setup. By substituting ΔL into Equation (Equation 11), the laser frequency fluctuations are estimated to be $3.47\times 10^{8}$ Hz/$\sqrt{\mathrm{Hz}}$ at 1 mHz.

### 3.3. Temperature Fluctuation Noise

Temperature fluctuations lead to OPD variations due to changes in the refractive index of the air, and thermo-elastic distortions of the optical elements and mounts. Temperature fluctuations usually limit the sensitivity of an interferometer in the frequency regime below 1 mHz [20,21]. Efforts such as inserting Viton strips and building Styrofoam boxes have been made to mitigate their effects to the operational environment during the experiment. To evaluate the residual thermal contributions to the differential measurements, ambient temperature is monitored by a temperature sensor attached to the outer wall of the vacuum chamber.

In contrast to the two noise sources introduced above, thermal fluctuation effects are recorded with a delay in the optical readout. In other words, the group phase between the temperature measurement and the actual effect on the differential interferometer must be considered in the analysis. Therefore, when analyzing temperature fluctuations, a spectral analysis method [20] is adopted instead of a time-domain linear fit.

Before applying the spectral analysis method, we first evaluate the correlation between the displacement and the temperature measurement by calculating the coherence between them. Figure 10 shows the amplitude and phase of the calculated coherence, and we observe a strong correlation (amplitude larger than 0.5) at frequencies below 1 mHz, while a non-zero phase is observed in the same frequency bandwidth. In our case, this shows that the major limitation presented by temperature fluctuations to our interferometer performance occurs in the sub-mHz regime and has a delay effect on the displacement measurement.

In this spectral analysis method, the transfer function between the temperature and the differential measurement is estimated by the equation
(14)$$H\left(\omega \right)=\frac{S_{T\phi }\left(\omega \right)}{S_{\mathrm{TT}}\left(\omega \right)},$$
where $S_{T\phi }\left(\omega \right)$ is the cross power spectral density of temperature to differential phase, and $S_{\mathrm{TT}}\left(\omega \right)$ is the power spectral density of the temperature measurement. Once the transfer function is determined, the contribution of the temperature fluctuation can be removed from the differential measurement in the spectral domain following
(15)$$\Delta \phi_{\mathrm{tempcorr}}\left(\omega \right)=\Delta \phi \left(\omega \right)-H\left(\omega \right)\;\cdot \;T\left(\omega \right).$$

Figure 11 shows the amplitude of the calculated transfer function between the displacement and the temperature measurement. From Figure 10, the correlation in the high frequency bandwidth above 1 mHz is negligible. Hence, directly applying the transfer function in Equation (Equation 15) will add noises in the uncorrelated frequency regime. A low-pass spectral filter is applied to the transfer function in order to remove its contribution, which is also plotted in Figure 11. The cut-off frequency of this low-pass filter is set to be 1 mHz based on the coherence results.

Figure 12 shows the logarithmic averaged LSD of the OPD-noise-corrected differential phase and the temperature correction results based on the spectral analysis method described in Equations (Equation 14) and (Equation 15). Temperature fluctuations clearly dominate in the low frequency regime below 0.5 mHz. After subtracting this noise contribution, the noise floor at 0.1 mHz is reduced from $6.5\times 10^{-9}$ m/$\sqrt{\mathrm{Hz}}$ to $9.8\times 10^{-11}$ m/$\sqrt{\mathrm{Hz}}$.

### 3.4. Detection System Noise Limit

The photodetector (PD) technical noise determines the minimum noise floor that an interferometric system can achieve, aside from fundamental limits such as shot noise. In a PD system, typical noise sources include dark current noise from the photodiode, shot noise originating from potential barriers, Johnson noise from thermal fluctuation in the resistor of the sensor circuits, and capacitive noise from the photodiode and circuit [29]. While other noise sources can be reduced by optimizing the circuit design [19], shot noise depends exclusively on the optical power received by the PD and its responsivity. In a heterodyne interferometer with a PLL phase readout, the minimum detectable displacement $d_{\min}$ can be derived when the signal-to-noise ratio (SNR) of the system is equal to 1 [30].

Assuming a small displacement that can be modeled as $A_{d}\sin\left(\omega_{d}t\right)$, the electric field of the interference signal in a general case, is expressed by
(16)$$\begin{array}{c} \begin{array}{c} E_{D}=E_{1}+E_{2}=A_{1}\exp\left[i\left(\omega_{1}t+\phi_{1}\right)\right]+A_{2}\exp\left[i\left(\omega_{2}t+\phi_{2}+A_{d}\sin\left(\omega_{d}t\right)\right)\right], \end{array} \end{array}$$
where $A_{i}$ are the amplitudes of the electric fields, $\omega_{i}$ are the optical frequencies, and $\phi_{i}$ are the phase terms of the individual beams. The irradiance on PD then can be represented as
(17)$$\begin{array}{c} \begin{array}{c} P_{D}=|E_{1}+E_{2}{|}^{2}=|A_{1}{|}^{2}+{\left|A_{2}\right|}^{2}+2A_{1}A_{2}\cos\left[\Delta \omega t+\Delta \phi +A_{d}\sin\left(\omega_{d}t\right)\right]. \end{array} \end{array}$$

By applying trignometric identities and Bessel expansion to Equation Equation 17, the irradiance on PD is rewritten as
(18)$$\begin{array}{c} \begin{array}{c} \begin{array}{cc} P_{D} & =P_{1}+P_{2}+2\sqrt{P_{1}P_{2}}\{J_{0}\left(A_{d}\right)\cos\left(\Delta \omega t+\Delta \phi \right)\\ \; & +\sum_{m=1}^{\infty }J_{2m}\left(A_{d}\right)\left[\cos\left(2m\omega_{d}t+\Delta \omega t+\Delta \phi \right)+\cos\left(2m\omega_{d}t-\Delta \omega t-\Delta \phi \right)\right]\\ \; & -\sum_{n=0}^{\infty }J_{2n+1}\left(A_{d}\right)\left\{\cos\left[\left(2n+1\right)\omega_{d}t+\Delta \omega t+\Delta \phi \right]-\cos\left[\left(2n+1\right)\omega_{d}t-\Delta \omega t-\Delta \phi \right]\right\}\}, \end{array} \end{array} \end{array}$$
where $P_{i}$ is the optical power in each inteferometer arm, $J_{\alpha }\left(x\right)$ is the Bessel function of the first kind. In this case, only small displacement is considered where $A_{d}\ll 1$. Based on the small argument approximation for Bessel function where
(19)$$\begin{array}{c} \begin{array}{c} J_{0}\left(x\right)=1,J_{1}\left(x\right)=\frac{x}{2},J_{2}\left(x\right)=\frac{1}{2}\frac{x^{2}}{2}, \end{array} \end{array}$$

Equation (Equation 19) can be simplified as
(20)$$\begin{array}{c} \begin{array}{c} \begin{array}{cc} P_{D} & =P_{1}+P_{2}+2\sqrt{P_{2}P_{2}}\{\cos\left(\Delta \omega t+\Delta \phi \right)\\ \; & -\frac{A_{d}}{2}\left[\cos\left(\omega_{d}t+\Delta \omega t+\Delta \phi \right)-\cos\left(\omega_{d}t-\Delta \omega t-\Delta \phi \right)\right]+O\left(A_{d}^{2}\right)\}. \end{array} \end{array} \end{array}$$

Therefore, the time average of the irradiance and the time-averaged incoherent square sum of the irradiance on the photodetector are
(21)$$\begin{array}{c} \begin{array}{c} \begin{array}{cc} \; & \langle P_{D}\rangle =P_{1}+P_{2},\\ \; & \langle P_{D}^{2}\rangle ={\left(P_{1}+P_{2}\right)}^{2}+P_{1}P_{2}A_{d}^{2}, \end{array} \end{array} \end{array}$$

The actual displacement to be measured, *d*, is converted by the relation $d=\lambda \;\cdot \;A_{d}/\left(2\pi \;\cdot \;2\right)$. The PD system noise in this case can be decomposed to shot noise based on analytical calculation, and other noises based on experimental measurements, following the expression
(22)$$\begin{array}{c} \begin{array}{c} \langle i_{\mathrm{PD}}^{2}\rangle =\langle i_{\mathrm{shot}}^{2}\rangle +\langle i_{\mathrm{other}}^{2}\rangle =2e\rho \langle P_{D}\rangle B+\frac{V_{M}^{2}}{G^{2}}, \end{array} \end{array}$$
where ρ is the detector responsivity, *B* is the detector bandwidth, $V_{M}$ is the measured output voltage from PD without incident optical signal, and *G* is the transimpedance gain and is usually specified by the manufacturer. Assuming the interferometer has a visibility of *V*, the SNR when PD noises are present can be expressed as
(23)$$\begin{array}{c} \begin{array}{c} \mathrm{SNR}=\sqrt{\langle I_{D}^{2}\rangle /\langle i_{\mathrm{PD}}^{2}\rangle }=k_{0}d\sqrt{\frac{4\rho^{2}P_{1}P_{2}}{2e\rho P_{0}B+\langle V_{M}^{2}\rangle /G^{2}}}, \end{array} \end{array}$$
where $k_{0}$ is the wavenumber. When SNR is equal to 1, the minimum detectable displacement is calculated as
(24)$$d_{\min}=\frac{\sqrt{2e\rho P_{0}+\langle V_{M}^{2}\rangle /\left(G^{2}\;\cdot \;B\right)}}{k\rho P_{0}V}\left[m/\sqrt{\mathrm{Hz}}\right]\;\cdot \;\sqrt{B}.$$

For the benchtop system, the RMS of output voltage square without detecting signal is $\langle V_{M}^{2}\rangle =1.29\times 10^{-7}{\left[V\right]}^{2}$, the optical power on the detector is $4.5\times 10^{-5}$ W, and the interferometer visibility is 0.45. By substituting the numerical values into Equation (Equation 24), the minimum detectable displacement is calculated to be $d_{\min}=$ $1.39\times 10^{-12}$ m/$\sqrt{\mathrm{Hz}}$.

### 3.5. Discussions

We investigated three common noise sources for heterodyne interferometry, including non-linear OPD noise, laser frequency noise, and temperature noise. The correction method for each noise source is identified, and the overall post-processing correction algorithm is described in the flow chart in Figure 13.

According to this correction algorithm, Figure 14 shows the differential measurement results (blue trace), the overall noise correction results (red trace), and the contribution of each noise source. The residual noise floor after noise correction shown in Figure 14 is $3.31\times 10^{-11}$ m/$\sqrt{\mathrm{Hz}}$ at 100 mHz. It is worth mentioning that this algorithm applies to uncorrelated noise sources in general heterodyne interferometry. If there are correlated noise sources, for example as in the DIFO readout and the temperature readout, the data needs to be pre-processed, before applying the correction algorithm to determine the frequency range of their correlation.

This post-processing noise correction algorithm provides an effective alternative to improve the sensitivity of heterodyne interferometers without implementing active stabilizations and can achieve high sensitivity over long-term measurements. Moreover, the application of the post-processing algorithm allows us to maintain an overall compact system footprint, and minimal complexity to the control system. The proposed interferometer and noise correction algorithm achieve a performance that is comparable to a few other techniques in GW applications, and has its own relative advantages, such as low hardware complexity and simple interferometer layout.

In addition to the benchtop system we built in the laboratory, this noise correction algorithm can be applied to general heterodyne interferometers [7,9]. Furthermore, this algorithm makes it possible to improve the performance of existing heterodyne optical readout systems without changing their configurations.

## 4. Conclusions and Outlook

We have presented a noise identification and noise correction algorithm that applies to general heterodyne interferometry. A compact heterodyne interferometer has been developed and tested to determine its performance, when applying this algorithm to its measurement results. We have developed a mitigation strategy for typical noise sources that are present in heterodyne interferometers such as non-linear OPD noise, laser frequency noise, and temperature fluctuations. The contribution from each noise source is estimated and subtracted from the main phase measurement. After correction, the noise floor is reduced by an order of magnitude to a level of approximately $3.31\times 10^{-11}$ m/$\sqrt{\mathrm{Hz}}$ at 100 mHz.

There are a few modifications to be made in the future to further improve the interferometer performance over the bandwidth above 1 mHz. On the operation environment side, the natural frequency of the pendulum platform can be reduced below 100 mHz to provide noise attenuation to a broader frequency band. To mitigate the non-linear OPD noise, better management of the crosstalk between RF signals can be implemented, such as building electromagnetic shield for each individual AOFS RF amplifier to reduce the amplitude of the sideband interference. On the instrument side, the interferometer design can be optimized to minimize the common-path noise term, $\phi_{n}$ in Equation (Equation 5), in the individual interferometers MIFO and RIFO. In addition, a laser source with a higher optical output power will reduce the shot noise level and, therefore, the overall interferometer noise floor at higher frequencies. Lastly, it will be interesting to investigate possible implementations of this algorithm using machine learning techniques and predicting filters to achieve a real-time operation and real-time sensitivity enhancement to laser interferometers without adding complex hardware for active stabilizations.

## Figures and Tables

**Figure 1 sensors-21-05788-f001:**
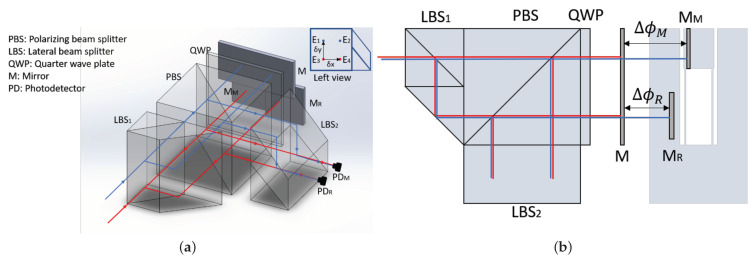
(**a**) The layout of the compact benchtop heterodyne laser interferometer design; (**b**) top view of the layout. Two laser beams with different frequencies enter the lateral beam splitter (LBS) and split into four beams, constructing two interferometers measuring interferometer (MIFO) and reference interferometer (RIFO). The MIFO is to measure the optical pathlength difference (OPD), $\Delta \phi_{M}$ and the RIFO is to measure $\Delta \phi_{R}$. The actual displacement of test mass M${}_{M}$ is calculated from the differential measurement between MIFO and RIFO.

**Figure 2 sensors-21-05788-f002:**
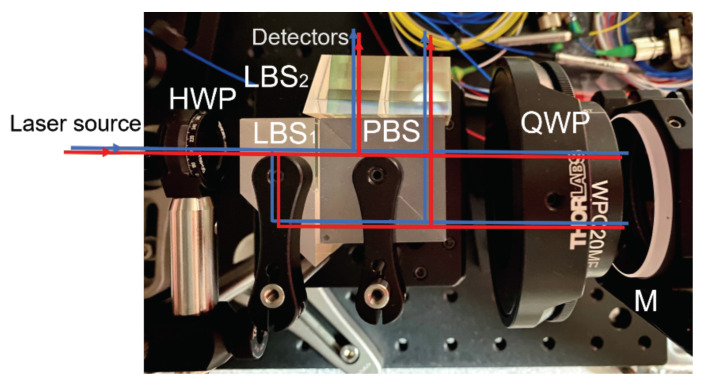
Layout of the benchtop compact heterodyne laser interferometer based on the design depicted in Figure 1. Only one static mirror M is used to characterize the system’s noise floor. Commercial optical and mechanical components are used to construct the benchtop system.

**Figure 3 sensors-21-05788-f003:**
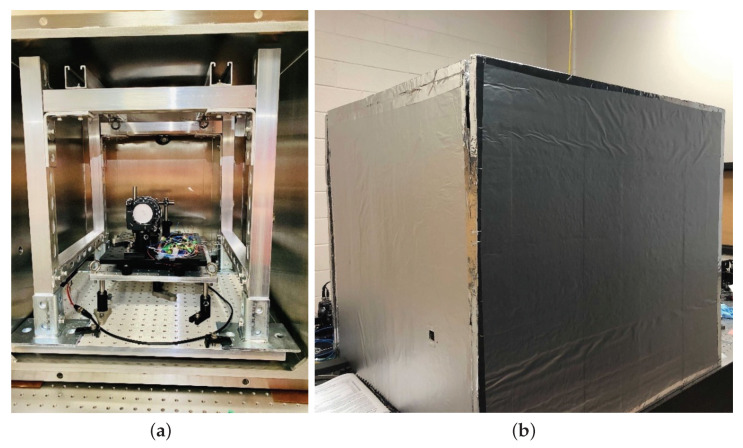
Operation environments. (**a**) Pendulum platform inside the chamber; (**b**) styrofoam thermal insulator outside the chamber. The compact interferometer is placed on a pendulum stage consisting of a breadboard and four steel wires. Viton strips are applied between the pendulum frame and the vacuum chamber to reduce ambient thermal coupling. The entire pendulum structure is fixed inside the vacuum chamber.

**Figure 4 sensors-21-05788-f004:**
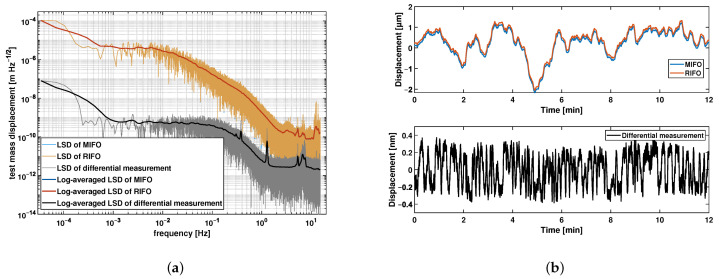
Preliminary test results. (**a**) LSD and its logarithmic average of the MIFO, RIFO, and differential measurements, respectively; (**b**) time series of a 12-min section of the MIFO, RIFO, and differential measurements. The traces of measurement results from MIFO and RIFO highly overlap in (**a**). Experimental results show a noise floor of $1.64\times 10^{-7}$ m/$\sqrt{\mathrm{Hz}}$ at 100 mHz for individual interferometers, and a noise floor of $2.76\times 10^{-10}$ m/$\sqrt{\mathrm{Hz}}$ at 100 mHz for the differential measurement, which is enhanced by three orders of magnitude from individual interferometers. The drift for differential measurement is $7.56\times 10^{-10}$ m over a period of 12 min.

**Figure 5 sensors-21-05788-f005:**
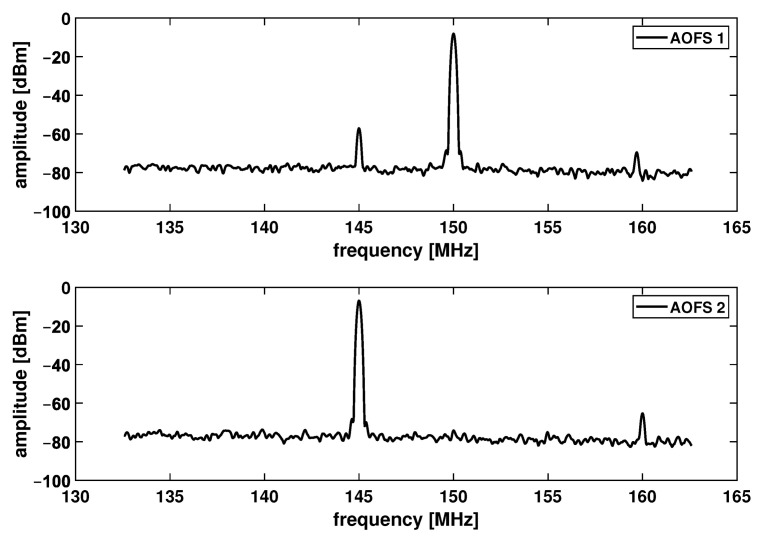
Spectra of RF driving signals for both AOFS. The frequency interval between this sideband and the main peak equals to the heterodyne frequency, leading to a ghost signal of unstable phase responsible for the non-linear OPD noise.

**Figure 6 sensors-21-05788-f006:**
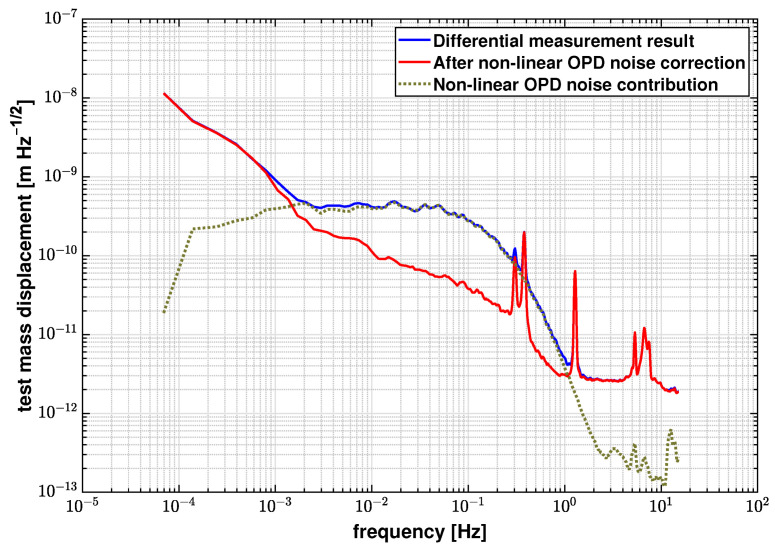
LSD logarithmic average of original differential measurements and the results after noise correction, and the OPD noise contribution. The noise floor is reduced from $2.76\times 10^{-10}$ m/$\sqrt{\mathrm{Hz}}$ to $3.86\times 10^{-11}$ m/$\sqrt{\mathrm{Hz}}$ at 100 mHz after applying the noise correction algorithm.

**Figure 7 sensors-21-05788-f007:**
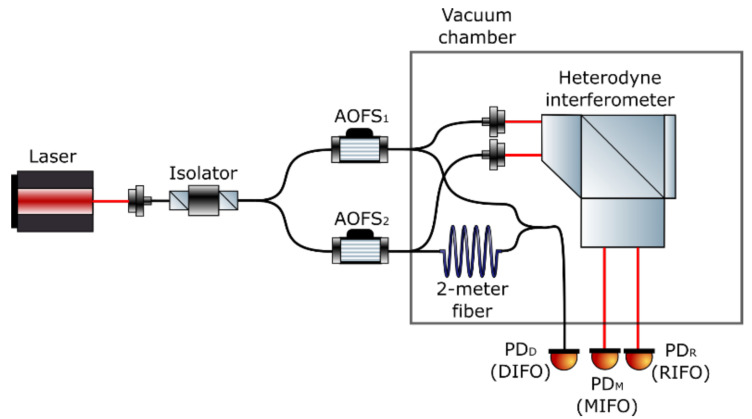
System layout with the delay-line interferometer integrated inside the vacuum chamber. The delay-line interferometer is constructed by inserting a 2-m fiber in one arm of the fiber Mach–Zehnder interferometer to amplify the effects of laser frequency noise.

**Figure 8 sensors-21-05788-f008:**
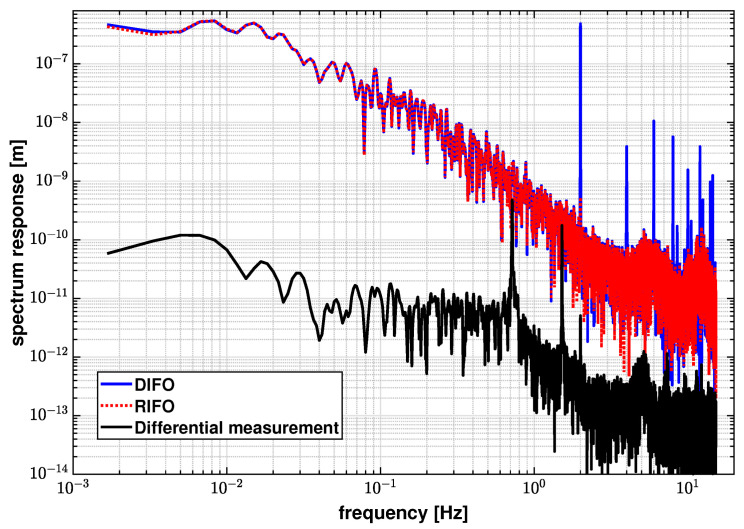
Linear spectra of DIFO, RIFO, and the differential measurement to the injected laser frequency noise by intentionally modulating the laser frequency at a rate of 2 Hz. The spectrum response of DIFO to the laser frequency noise is $4.82\times 10^{-7}$ m while the RIFO response is $5.30\times 10^{-10}$ m at 2 Hz. After performing the differential operation, the spectrum response is reduced to $5.07\times 10^{-12}$ m at 2 Hz.

**Figure 9 sensors-21-05788-f009:**
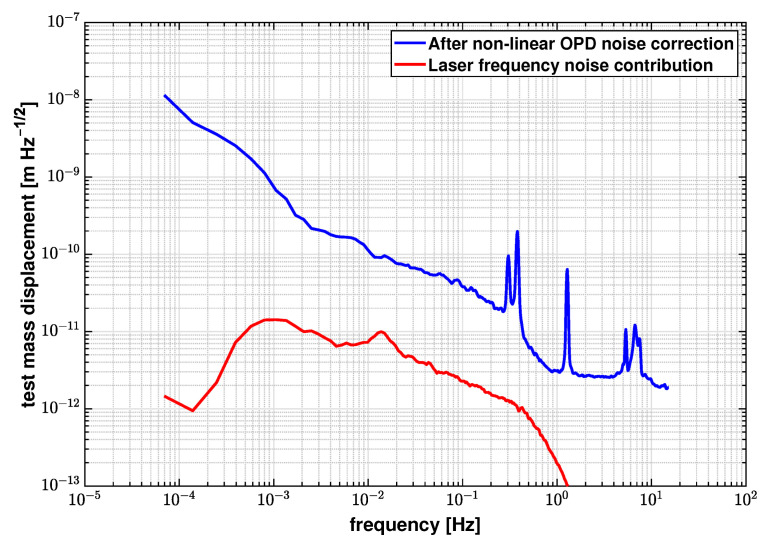
Laser frequency noise contribution estimated by a linear fit, compared to the OPD-noise-corrected differential measurement. The laser frequency noise is estimated to be $2.28\times 10^{-12}$ m/$\sqrt{\mathrm{Hz}}$ at 100 mHz, and is not the limiting noise source in our benchtop experiment.

**Figure 10 sensors-21-05788-f010:**
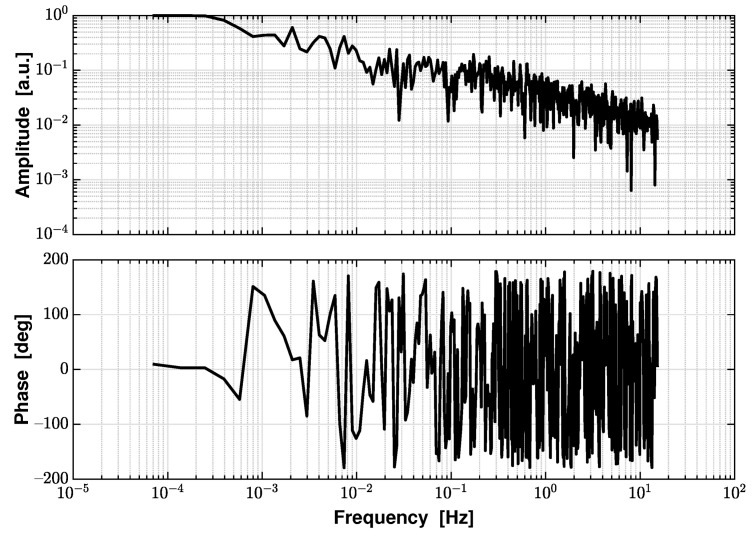
The amplitude and phase of the coherence function between the displacement and the temperature measurements. In the low frequency regime below 1 mHz, it shows a strong correlation between the displacement and the temperature measurement as the amplitude is large than 0.5, and a time delay effect as the phase is nonzero.

**Figure 11 sensors-21-05788-f011:**
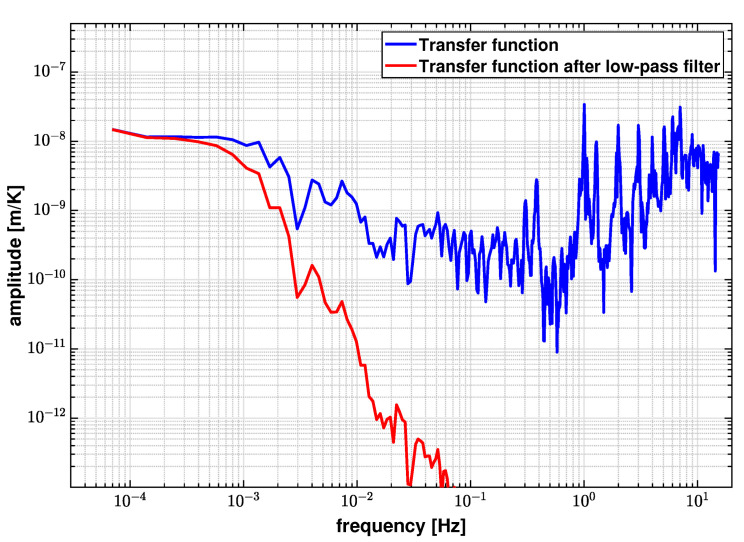
Transfer function between the displacement and temperature measurements before and after applying a 1 mHz low-pass filter. The amplitude of the transfer function is attenuated by a low-pass filter in the frequency regime where the correlation is negligible.

**Figure 12 sensors-21-05788-f012:**
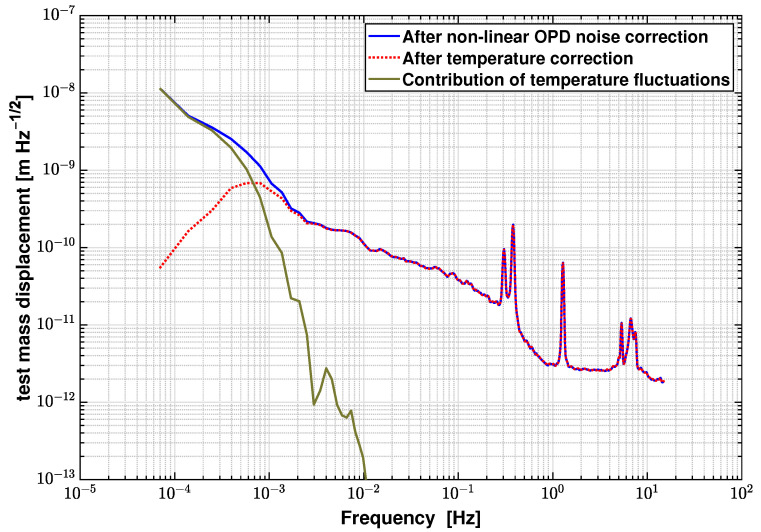
Contributions of temperature fluctuations estimated by a spectral analysis method, and the measurement results before and after correction of this noise source. The noise floor is reduced from $6.5\times 10^{-9}$ m/$\sqrt{\mathrm{Hz}}$ to $9.8\times 10^{-11}$ m/$\sqrt{\mathrm{Hz}}$ at 0.1 mHz after applying the noise correction algorithm.

**Figure 13 sensors-21-05788-f013:**
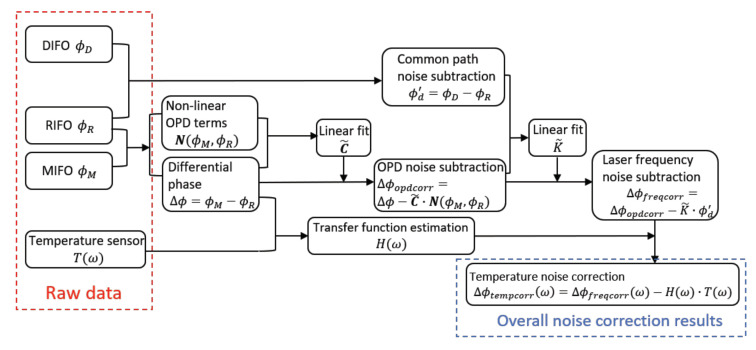
Flow chart of the overall noise correction algorithm. The non-linear OPD noise, laser frequency noise and the temperature fluctuation noise are characterized, and the contribution from each noise source is removed with this algorithm.

**Figure 14 sensors-21-05788-f014:**
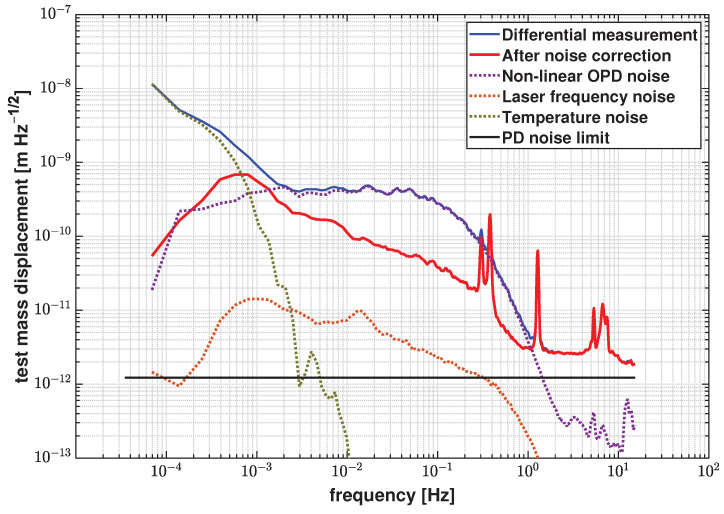
Overall noise correction results based on the differential measurement and contribution of each noise source. The residual noise floor is $3.31\times 10^{-11}$ m/$\sqrt{\mathrm{Hz}}$ at 100 mHz, and $9.8\times 10^{-11}$ m/$\sqrt{\mathrm{Hz}}$ at 0.1 mHz after correction. The detection system noise level is $1.39\times 10^{-12}$ m/$\sqrt{\mathrm{Hz}}$ and also marked in this plot.

**Table 1 sensors-21-05788-t001:** Linear fitting coefficients and estimated fitting errors.

*i*	1	2	3	4
$C_{i}$	$3.62\times 10^{-3}$	$1.11\times 10^{-3}$	$1.89\times 10^{-6}$	$-2.51\times 10^{-6}$
$\delta C_{i}$	$1.96\times 10^{-6}$	$1.96\times 10^{-6}$	$1.23\times 10^{-6}$	$1.24\times 10^{-6}$

## Data Availability

Not applicable.

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
