# Peer review of "Investigation and Mitigation of Noise Contributions in a Compact Heterodyne Interferometer"

_sensors, 2021, doi:10.3390/s21175788_

Round 1
Reviewer 1 Report
Authors are presenting a study on noise contributions on heterodyne compact interferometer for application from displacement measuring interferometry, laser interferometers in space to gravitational wave detection.
Presented measurement noise sources including shot noise, temperature noise and the laser noise are well described and substracted. Authors are showing an algorithm to substract the thermal noise from raw data of three uncorrelated interferometers with parallel measurement of temperature reducing the 100 mHz residual noise floor. Authors are presenting also the future improvements in methods to further improvement of the interferometer low pass noise.
I recommend this paper for publishing with few following minor changes.
The text is written with very good English and I didn´t find a particular problem here.
Anyway I have found some small mistakes in the descriptions and I have some possible recommendation for full understand the text.
- Line 83 – Authors are referring to Figure 2 (b) in the text but it looks that the correct reference is Figure 1 (b)
- Line 87-89 – I would recommend to write full explication before abbreviation MIFO and RIFO as well as OPD and LBS in the Figure caption
- I found missing units in Amplitude (y-axis) on the top plot in Figure 10.
Reviewer 2 Report
In this paper the authors present a compact interferometer made up of several components that are bonded together. The interferometric technique of choice is heterodyne interferometry, where a pair of acousto-optic modulators (AOM) are used to frequency-shift a pair of beams from the same laser source, leading to a heterodyne signal of 5 MHz in the interfered field. The input beam pair passes through an initial beam splitter that produces four beams, two for a reference interferometer (RIFO) and two for a measurement interferometer (MIFO). MIFO can be used to measure the displacement of a test mass in one degree of freedom, and RIFO can be used to subtract common-mode noise from this measurement.
After this initial beam splitter, a polarizing beam splitter in combination with a quarter waveplate are used to retro-reflect these beams off of a set of mirrors. The beams on the interferometers' static arms reflect on a fixed mirror, whereas the beams on the interferometers' dynamic arms reflect on the reference mirror (for RIFO) and the target mirror (for MIFO). The light is then combined with the help of a third beam splitter, and the resulting fields are captured by a pair of photoreceivers. In the initial implementation of this device, the fixed mirror, as well as the reference and target mirrors, are the same mirror.
The authors then present a strategy to mitigate some of the noise present in the differential measurement between MIFO and RIFO. First, two linear fit methods are introduced. The first one deals with the resulting phase noise from the cross coupling of the high power RF signals used to drive the AOMs. The second one deals with the residual laser frequency noise, which is measured with the help of a third interferometer (DIFO) that features an intentional arm length difference of 2 meters so that frequency noise becomes dominant. Then, a transfer function of temperature to differential phase fluctuations is determined, and this noise is removed at low frequencies (using a 1 mHz low pass filter, a decision informed by the measured coherence function between the displacement and temperature measurements) in the spectral analysis. Lastly, the authors include a calculation of the fundamental limit to displacement sensitivity imposed by the interferometer's shot noise.
The paper reads well and it is well referenced. The results are clearly presented and the conclusions drawn from them are sound. While the noise mitigation methods are not a novelty, the paper offers a nice compendium, and it will be a useful read to the many people involved with high precision interferometry.
Below are my remarks to improve the paper.
REMARK 1: Figure 5 seems to show the spectra of the RF signals driving the pair of AOMs, showing spurious sidebands spaced by multiples of the heterodyne frequency from the carrier. However, the text and figure caption suggest that this is the “measured spectra of the laser beams after passing through the AOFS”. This does not make sense to me.
REMARK 2: It is not trivial how one obtains Equation 16. The time average implies integrating a period of \cos (\omega_h t + A_d*\sin(\omega_d t)), which does not seem trivial. In addition, power squared leads to amplitude to the power of 4 (P^2 ~ E^4 ~ A^4), therefore there seems to be a typo after the first equal sign (i.e., |E_D|^2 instead of |E_D|^4). Please provide better clarification of this equation.
REMARK 3: Nothing is said about the bonding technique used to build the interferometer. It would be good if the authors could explain briefly how this was achieved. Of particular interest is the potential impact that the bonding has on the AR coatings that are likely needed and present in some of the contacted surfaces (e.g., on the QWP’s back surface).
REMARK 4: It seems that in Figure 1, the blue and red beams from the left drawing correspond to the red and blue beams from the right drawing. It would be good to have matching colors on both drawings.
REMARK 5: All of the included graphics seem to be bitmap images. I would like to encourage the authors to replace them with vector graphics.
REMARK 6: I suggest the authors to change all instances of $var_{some text}$ to $var_{\text{some text}}$ for a more elegant presentation.
REMARK 7: The way the interferometers’ structural noise is described is somewhat confusing (equations 1-4, and equations in lines 94 and 96). This is what I understand: \phi_n(x,y) describes the noise in a some reference plane (perhaps at the plane defined by mirror M?), as probed by any of the four beams, and \phi_n(0,0) (i.e., the noise probed by field E_3, which is the static arm field of the MIFO) is taken as reference to be zero (i.e., as the interferometer deforms, the phase of E_3 at the reference plane remains unchanged). Then, \Delta \phi_R and \Delta \phi_M represent the additional contribution to the noise due to propagation from the reference plane to M_R and M_M respectively in the interferometers’ dynamic arms. It follows that the interfering fields show a dependency only on the \delta y structural stability of the interferometer, and not on \delta x (this conclusion is self-evident through inspection of the system). It would improve the paper to make this analysis more clear in the text and drawings. In particular, the authors should elaborate on the definition of \phi_n(x,y).
